# High Use of Dietary Supplements and Low Adherence to the Mediterranean Diet Among Italian Adolescents: Results from the EduALI Project

**DOI:** 10.3390/nu17132213

**Published:** 2025-07-03

**Authors:** Sofia Lotti, Marta Tristan Asensi, Donato Cretì, Erika Mollo, Armando Sarti, Francesco Sciattella, Monica Dinu, Barbara Colombini, Luigi Rizzo, Francesco Sofi

**Affiliations:** 1Department of Experimental and Clinical Medicine, University of Florence, 50134 Florence, Italy; sofia.lotti@unifi.it (S.L.); monica.dinu@unifi.it (M.D.); barbara.colombini@unifi.it (B.C.); francesco.sofi@unifi.it (F.S.); 2Valori Aziendali—Mediterranean Food Science, 50141 Florence, Italy; d.creti@valoriaziendali.it (D.C.); info@dietistaerikamollo.it (E.M.); francesco.sciattella10@gmail.com (F.S.); l.rizzo@valoriaziendali.com (L.R.)

**Keywords:** adolescents, dietary supplements, Mediterranean diet, eating habits

## Abstract

**Objective:** Eating habits established during adolescence play a crucial role in shaping both current and future health status. However, adolescents today appear to be increasingly adopting poorer dietary and lifestyle habits. This study aims to investigate eating behaviors and supplement use among adolescent students who participated in the EduALI nutrition education project. **Methods:** The project included first-year students from six sports-oriented scientific high schools in Florence. The students participated in six nutrition sessions, completing the International Physical Activity Questionnaire to assess physical activity performance, a questionnaire on dietary supplement use, and, to assess eating habits a Food Frequency Questionnaire (FFQ), as well as the Medi-Lite questionnaire to specifically assess adherence to the Mediterranean diet (MD). The data from the FFQ were compared with national dietary guidelines. **Results:** The study sample consisted of 83 students (69.9% male; average age: 13.8 ± 0.4 y). The results showed that 61.4% of participants had used supplements in the past year, most commonly mineral salts (56.6%) and vitamins (54.2%), followed by botanical products (22%), sports drinks (16%), proteins/amino acids (8%), and creatine (2%). Alarmingly, 27.7% used supplements without medical supervision. Students involved in individual sports had higher supplement consumption than those in team sports, especially creatine use. Eating habits revealed poor adherence to both the MD and Italian dietary guidelines, with deficiencies in fruits, vegetables, olive oil, fish, and legumes, and excessive intake of cheese, meat, especially red and processed meats. **Conclusions:** A high prevalence of supplement use among adolescents was observed, along with poor adherence to dietary guidelines and MD. These findings underscore the need for targeted, school- and sport-based interventions to enhance adolescents’ nutritional awareness and responsible supplement use.

## 1. Introduction

Adolescence is a critical period for developing lifelong nutritional habits that shape both immediate health outcomes and long-term disease prevention [1]. During this stage, adolescents undergo rapid physical growth and hormonal fluctuations that significantly elevate their nutritional requirements and make them more susceptible to the effects of dietary patterns. These biological changes are accompanied by a growing sense of independence, which often leads to greater autonomy in food choices [2]. However, without proper nutritional education and guidance, this autonomy can result in poor eating behaviors—such as increased consumption of ultra-processed foods, irregular meal patterns, and inadequate intake of essential nutrients [3]. These habits, once established, tend to persist into adulthood and are strongly associated with an increased risk of chronic health conditions, including obesity, type 2 diabetes, and cardiovascular disease [4,5].

Unfortunately, many adolescents adopt unhealthy eating behaviors, such as skipping meals, consuming fast food, and choosing snacks high in sugar and fats, while neglecting essential food groups such as fruits, vegetables, and whole grains [6,7]. Our previous research found that Italian adolescents exhibit only moderate adherence to the Mediterranean diet (MD), with a high consumption of meat and processed products and insufficient intake of vegetables, legumes, and fish [8]. These dietary patterns impact both short-term health and long-term disease prevention [9].

In addition to poor dietary habits, there is growing concern over the increasing use of dietary supplements among adolescents. Defined as sources of nutrients or other substances intended to complement the regular diet [10], supplements are used by 26% of U.S. adolescents aged 14–18 [11] and by 35% of adolescents in Italy [12]. Dangerously, use is often not medically supervised but driven by peer influence, social media, or celebrity endorsements, raising concerns about the understanding of proper nutrition and the potential health risks associated with unregulated use [13].

This study aims to investigate dietary supplement use and examine eating behaviors in adolescent students by comparing the consumption of specific food groups with national dietary guidelines, with a focus on adherence to the MD. The findings will contribute to the limited evidence on this topic and support educational efforts to promote healthier eating and more informed supplement use.

## 2. Materials and Methods

### 2.1. Study Design and Participants

This research is part of the “EduALI Project”, promoted by the Department of Experimental and Clinical Medicine of the University of Florence and the Valori Aziendali Training Agency—Mediterranean Food Science. This study involved first-year classes from six sports-oriented scientific high schools in the province of Florence, Italy. During 2024, students attended six lessons on food education. The project aimed to raise awareness about eating habits by addressing health, environmental, economic, and social aspects, using methods engaging for adolescents. Topics included healthy eating patterns, sensory analysis of food, sustainability, and food safety. At the end of each lesson, the adolescents in the class were asked to complete questionnaires investigating different aspects of their lifestyle, including dietary behaviors, physical activity, and the use of supplements (Appendix A). The questionnaires were self-administered to all the participants, who answered them anonymously. Each session lasted one hour, with 40 min of lecture and 20 min for questionnaire completion.

After the questionnaires were completed, the investigators entered the anonymous data directly into an electronic datasheet, with each participant identified by a unique alphanumeric ID.

This study was conducted according to the guidelines of the Declaration of Helsinki and was approved by the Ethics Committee of the University of Florence, Florence, Italy (n. 1261/2020, protocol 0161824).

### 2.2. Data Collection

#### 2.2.1. Sociodemographic and Anthropometric Parameters

Data on socio-demographic (sex, age, number of people in the family, having brothers or sisters, and type of diet followed) and anthropometric (weight and height) variables were collected. Using the height and weight reported by participants, the body mass index (BMI) was calculated.

#### 2.2.2. Physical Activity

Participants’ physical activity was assessed using the long version of the International Physical Activity Questionnaire (IPAQ), which evaluates the level of physical activity in four domains: transport (moving from place to place), work, household chores and leisure time, including moderate and vigorous activity, and walking during the previous week. The IPAQ has been used to estimate physical activity levels and has been shown to be reliable in several contexts, including in adolescents [14,15]. The results are expressed in MET-minutes/week, calculated by multiplying the MET value of each activity (vigorous-8 MET, moderate-4 MET, walking-3.3 MET) by the number of days the activity was performed during the week. From the total score, the physical activity level is divided into three categories: low, moderate, and high.

#### 2.2.3. Use of Dietary Supplements

To assess dietary supplement use, a comprehensive questionnaire was administered, developed based on previous studies of Italian students [12,16]. The first section includes three questions aimed at determining whether the participant uses supplements, whether they do so for health-related reasons, and whether they are following medical advice when supplementing their diet. Respondents could answer: “Yes”, “No” or “Don’t know.” Supplement users were then asked to specify which supplements they take by selecting from a table with broad categories (vitamins, herbs or botanicals, mineral salts, or other supplements), each with detailed options. The second section investigates sources of information and trust regarding supplements, including books, supplement labels, family/friends, health food stores, media, pharmacists, physicians, the internet, or others.

#### 2.2.4. Eating Habits

Participants’ eating habits were assessed using a medium-length food frequency questionnaire (FFQ), validated in the Italian population [17]. The questionnaire asked participants to report 36 commonly consumed foods and corresponding portion sizes. They also indicated habitual frequency consumption, choosing from seven categories, ranging from “never” or “less than once a week” to “seven times a week”. Foods were categorized into beverages, milk and dairy products, meat, fish, eggs, cereals, vegetables, legumes, fruits, fatty condiments, and others (sweets, fried foods, and fast food). For beverages such as coffee, alcohol, and soft drinks, quantities were given in cups or glasses.

Adherence to the MD was assessed using the validated Medi-Lite questionnaire [18], which includes nine items that assess daily intake of fruits, vegetables, cereals, meat and meat products, dairy products, alcohol, and olive oil, as well as weekly intake of legumes and fish. For foods typical of the MD (fruits and vegetables, cereals, legumes, and fish), 2 points are assigned for the highest consumption category, 1 point for the middle category, and 0 points for the lowest category. For olive oil, 2 points are assigned for regular use, 1 point for frequent use, and 0 points for occasional use. Foods not typical of the MD (meat and meat products, dairy products) are scored as follows: 2 points for the lowest consumption category, 1 point for the middle category, and 0 points for the highest category. In this study, the alcohol consumption item was excluded from the Medi-Lite questionnaire due to the young age of the participants, consistent with previous research conducted in adolescent populations [8]. Thus, the final score ranged from 0 (low adherence) to 16 (high adherence).

### 2.3. Statistical Analysis

Only adolescents who completed all the required questionnaires were included in the final analysis. Since the lessons were conducted over five different days at each school, not all students were able to attend every session, resulting in some variability in participation.

Statistical analyses were performed using PASW 27.0 for Macintosh (SPSS Inc., Chicago, IL, USA). The continuous variables were expressed as means ± standard deviations (SDs), while the categorical variables were presented as frequencies and percentages (%). Group comparisons for continuous variables were assessed using the Mann–Whitney or Kruskal–Wallis tests, depending on the number of groups, while the Chi-square test was employed to analyze categorical variables. A *p*-value < 0.05 was considered statistically significant for all analyses.

Additionally, the dietary intake for each food group, as reported in the FFQ, was compared with the current Italian dietary guidelines for adolescents aged 11 to 14 years [19]. Sub-analyses were also performed based on the type of sport practiced (team vs. individual).

## 3. Results

The sample included a total of 83 high school students with an average age of 13.8 ± 0.4 years, 69.9% (n = 58) of whom were boys. The mean BMI was 20.0 ± 1.8 kg/m^2^, with 19.3% (n = 16) of the students classified as underweight, 78.3% (n = 65) as having a normal weight, and 2.4% (n = 2) as overweight or obese. None reported eating a vegetarian or vegan diet. A large percentage of the sample reported having siblings (85.5%) and living with three (47%) or four people (26.5%) in the household.

An analysis of the IPAQ questionnaire revealed a total score of 6407.3 ± 4814.7, indicating a high level of physical activity overall. Specifically, 86% (n = 71) of the participants were “very active”, while a smaller group was classified as “sufficiently active” (n = 11, 13.3%), and only one participant was “sedentary” (1.2%). These results are not surprising, considering that the students involved in this study attended a sports school. No significant differences in physical activity levels were found between boys and girls (*p* = 0.812).

### 3.1. Use of Dietary Supplements

Regarding the use of supplements, 61.4% of the students had consumed some type of supplements in the past year, and only 27.7% of the students stated that their supplement use was based on medical advice.

Analyzing the consumption by type of supplement, it emerged that more than 50% of the sample reported consuming mineral salts (n = 47, 56.6%) or vitamins (n = 45, 54.2%), with magnesium (n = 39, 47%), potassium (n = 39, 47%), vitamin D (n = 19, 22.9%), and vitamin C (n = 37, 44.6%) among the most commonly used. About 20% reported also using botanicals. Sports-related supplements, such as sports drinks (n = 13, 15.7%), protein (n = 7, 8.4%), and creatine (n = 2, 2.4%), as well as other supplements, such as melatonin (n = 6, 7.2%), commonly used to regulate sleep, were reported to a lesser extent (Figure 1).

Students cited doctors as their most trusted source of information on supplement use (75.9%), followed by product labels (39.8%), family members (38.6%), and pharmacists (32.5%). Fewer students sought advice from the internet (12%) or friends (8.4%).

### 3.2. Eating Habits

By analyzing the eating habits reported through the FFQ and comparing them with the recommended intake values of the Italian guidelines for adolescents aged 11 to 14 years, it was found that the average consumption of most foods was not in line with the recommendations (Table 1). On the one hand, the total sample reported consumption of fruit (−2.2 times), vegetables (−4.1 times), olive oil (−3.7 times), fish (−2.5 times), and legumes (−1.9 times) well below the recommended levels. On the other hand, an overconsumption of cheese and meat was observed, particularly for red meat and sliced meat (+3.7 and +2.5 times).

Analyzing adherence to the MD, the average Medi-Lite score among students was 7.6 ± 2.2, indicating low adherence to the dietary model. The analysis of individual Medi-Lite items showed that over 20% of the adolescents did not consume at least one serving of fruit (n = 18, 21.7%) or vegetables (n = 20, 24.1%) per day, and about half of the adolescents did not consume at least one serving of legumes (n = 40,48.2%) or fish (n = 42, 50.6%) per week.

### 3.3. Sub-Analysis Categorized by Type of Sport

To analyze possible differences in supplement use in relation to the type of sport played, the sample was divided between those who played team and individual sports. The “team sports group” included 55 students (66.3%) who played football, basketball, and volleyball. In contrast, the “individual sport group” included the remaining 28 students (33.7%) who practiced running, tennis, artistic gymnastics, and swimming.

A trend (*p* = 0.0736) suggested that adolescents involved in individual sports were more likely to report supplement use in the past year (71.4%) compared to those in team sports (56.4%). When specific supplements were analyzed, adolescents who played individual sport showed significantly higher creatine consumption (+7%, *p* = 0.045), while those who played team sport had a notably higher intake of calcium (+18%, *p* = 0.031). In contrast, the analysis of other minerals revealed an opposite pattern: adolescents in individual sports consumed more magnesium (+15.3%, *p* = 0.186) and potassium (+20.7%, *p* = 0.074), though these differences did not reach statistical significance. No significant differences in vitamin use were found between the two groups (Figure 2).

Finally, when analyzing the most reliable sources consulted for supplement use according to the type of sport practiced, no statistically significant differences emerged. However, a trend (*p* = 0.072) was observed in that adolescents participating in group sports (85.7%) seemed more likely to consult the doctor before taking supplements than those participating in individual sports (67.3%).

## 4. Discussion

The results of this study reveal concerning patterns in supplement use and dietary habits among Italian adolescent students. More than half of the participants reported using dietary supplements in the past year, with many doing so without medical supervision. In addition, our sample showed a low adherence to the MD and the Italian dietary guidelines for adolescents.

Nutrition plays a fundamental role in the growth, development, and long-term health of adolescents. Early adoption of healthy eating habits, in accordance with guidelines and associated with the MD, is crucial for reducing the risk of chronic diseases such as obesity and cardiovascular disease, as well as promoting healthy aging [4,20]. In particular, diet quality—which includes a balanced intake of macronutrients and micronutrients—plays a key role in overall health [21].

However, in our study, we observed incorrect eating habits in adolescent students that were far from both the MD and the dietary guidelines. Specifically, the consumption of fruits, vegetables, fish, and legumes was well below recommended levels, while meat and sliced meats were consumed excessively. Furthermore, we found that the intake of olive oil, which is the preferred fat in the MD, was insufficient in this group of adolescents, mirroring a similar trend observed in a previous study of ours [8]. These dietary imbalances are consistent with other research showing that adolescents often fail to meet nutritional recommendations, particularly in terms of nutrient quality [22,23].

The primary driver behind these imbalances is likely insufficient nutrition education [24]. Research has shown that a lack of adequate nutritional knowledge can lead to suboptimal food choices, such as excessive protein intake and insufficient consumption of other essential nutrients [25]. Furthermore, adolescents often lack personalized nutrition guidance due to limited access to qualified experts, which, combined with time constraints from sports, training, and academic commitments, exacerbates the challenge of maintaining a balanced diet [14,26]. These factors underscore the need for better educational strategies to improve adolescents’ understanding of nutrition’s role in overall health.

Another noteworthy finding is the high prevalence of dietary supplement use in this sample. More than half of the students reported using supplements in the past year, a rate significantly higher than the 25–35% typically reported in general adolescent populations [11,12]. This discrepancy may be attributable to the specific nature of our sample, which consisted solely of adolescent students attending a sports-focused high school. Previous studies have shown that adolescents engaged in organized sports or athletic programs are more likely to use dietary supplements compared to their non-athlete peers, primarily to enhance performance, recovery, or overall physical condition [27,28]. Notably, most of the supplements used in our study were sport-related—such as mineral salts, vitamins, sports drinks, protein/amino acids, and, to a lesser extent, creatine. These findings also align with existing literature reporting that sports-specific supplements are commonly used among adolescent athletes [29,30].

An interesting trend emerged within the data: adolescents involved in individual sports tended to use more supplements, particularly creatine, compared to their counterparts in team sports. This is consistent with the findings in the literature, which suggest that those who practice individual sports often adopt more aggressive nutritional and supplementation strategies to improve performance. In contrast, athletes in team sports tend to use supplements more conservatively [27]. This difference is likely due to the distinct priorities and psychological dynamics between the two groups. Individuals involved in individual sports are more focused on optimizing personal performance, while those in team sports prioritize collective success, which relies more on strategy, cohesion, and teamwork than on individual achievements [31].

In addition, a concerning aspect of our findings is that nearly one-third of adolescents who used supplements did so without medical supervision, exposing them to potential health risks. Without proper guidance, these adolescents could face nutrient imbalances, contamination or even consumption of banned substances. This aligns with literature showing that adolescents in sports often rely on sources other than healthcare professionals for supplement information, such as parents, coaches, teammates, or online platforms [28,30]. Unfortunately, many adolescents and their coaches often lack adequate knowledge about the potential adverse effects of the supplements they use [32,33].

This study has several limitations. One key factor is the relatively small and predominantly male sample, which may affect the generalizability of the findings. Additionally, reliance on self-reported questionnaires introduces another limitation, as participants’ responses may be influenced by biases related to accuracy and honesty. To mitigate this, research team members were present in the classroom to assist with clarifications and provide support during questionnaire completion. Moreover, data on weight and height were self-reported, which could introduce recall or reporting bias, particularly in the calculation of BMI. In addition, the consumption of certain food groups, such as fruits and vegetables, varies according to seasonality and may not be accurately captured when estimating annual intake using a FFQ. It is also important to acknowledge that potential confounding factors, such as socioeconomic status and dietary education at home, were not assessed in this study and may have influenced the results. Despite these limitations, this study also has notable strengths, such as the inclusion of multiple schools, which strengthens the findings by reducing the risk of selection bias. Furthermore, the analysis of various aspects of participants’ lifestyles—such as physical activity, dietary habits, and supplement use—offers a more comprehensive understanding of their overall health behaviors.

The high prevalence of supplement use among adolescent students, coupled with low adherence to dietary guidelines and the MD, warrants the attention of experts. These behaviors may lead to adverse health outcomes, such as nutrient imbalances, increased risk of chronic diseases, and harmful effects from the improper use of supplements. Given the widespread use of supplements and evident gaps in nutritional knowledge, targeted interventions, such as nutrition education programs in schools or sports centers, are urgently needed. Additionally, schools and sports organizations must take a more proactive role in providing accurate nutritional information and fostering an environment where health and nutrition are integral components of young people’s lives.

## Figures and Tables

**Figure 1 nutrients-17-02213-f001:**
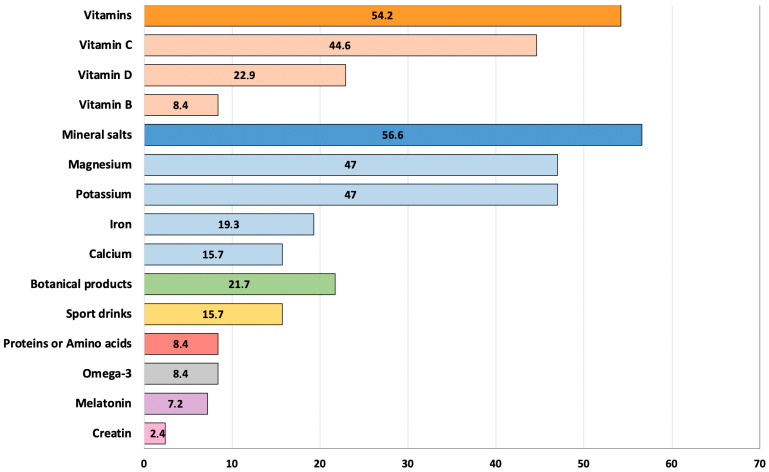
Use of food supplements in the total sample.

**Figure 2 nutrients-17-02213-f002:**
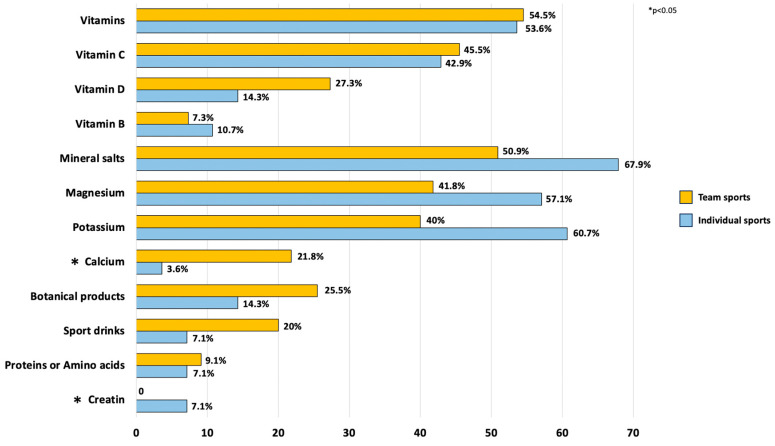
Use of food supplements according to the type of sport. * *p*-value < 0.05.

**Table 1 nutrients-17-02213-t001:** Comparison of food group consumption with the dietary recommendations outlined in the Italian guidelines, with red indicating overconsumption and yellow underconsumption.

	Recommended Frequency of Consumption	All (n = 83)
Red meat, g/week	0–100	367.0 ± 248.2
White meat, g/week	200–300	319.5 ± 246.4
Sliced meats, g/week	0–50	125.9 ± 113.5
Fish, g/week	450	180.1 ± 165.1
Legumes, g/week	360	191.6 ± 192.2
Cheeses, g/week	250	289.2 ± 167.6
Milk, g/day	200	93.6 ± 82.3
Yogurt, g/day	125	41.3 ± 44.5
Pasta or rice, g/day	100	99.7 ± 42.1
Bread, g/day	150	42.4 ± 31.2
Potatoes, g/week	400	316.6 ± 261.8
Vegetables, g/day	400	98.5 ± 78.8
Fresh fruit, g/day	240–360	109.7 ± 90.6
Olive oil, g/day	30	8.1 ± 5.5

## Data Availability

Additional data are available from the corresponding author on reasonable request.

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
