# Peer review of "High Use of Dietary Supplements and Low Adherence to the Mediterranean Diet Among Italian Adolescents: Results from the EduALI Project"

_nutrients, 2025, doi:10.3390/nu17132213_

Round 1

Reviewer 1 Report

Comments and Suggestions for Authors

This is a comminication article with adequate novelty. However, some points should be addressed.

  • In the Abstract, some background statements could help the readers to understand more easily the concept of the article.
  • In the Conclusions section of the Abstract, the authors should provide some proposals to treat the problem found in their study.
  • The 2nd sentence of the Introduction (lines 30-32) needs a bit more analysis.
  • It could be very useful for the readers to include as supplementary material the questionnaires investigating different aspects of  lifestyle, including dietary behaviors, physical activity and the use of supplements.
  • Please, report the normality distribution test that you used.
  • The authors should include in the limitations of the study that most of participants were boys and that BMI was calculated by self-reported data which may lead to recal bias.

Author Response

In the Abstract, some background statements could help the readers to understand more easily the concept of the article.

R: Thank you for the helpful suggestion. A background sentence has been added so that the reader can better understand the context (Please see lines 10-12).

In the Conclusions section of the Abstract, the authors should provide some proposals to treat the problem found in their study.

R: Thank you for the suggestion. A sentence with some proposal to treat the problem found in our study has been added in the conclusion as requested (Please see lines 28-30).

The 2nd sentence of the Introduction (Please see lines 30-32) needs a bit more analysis.

R: Thanks for the suggestion. The second sentence of the introduction has been deepened as requested (Please see lines 35-59).

It could be very useful for the readers to include as supplementary material the questionnaires investigating different aspects of lifestyle, including dietary behaviors, physical activity and the use of supplements.

R: Thank you for the helpful enquiry. We have added the questionnaires used as supplementary material 1.

Please, report the normality distribution test that you used.

R: Thank you for the observation. Since we applied non-parametric statistical tests in our analysis (e.g., Mann-Whitney, Kruskal-Wallis), we did not perform a normality test, as these methods do not assume a normal distribution.

The authors should include in the limitations of the study that most of participants were boys and that BMI was calculated by self-reported data which may lead to recal bias.

R: Thank you for the suggestion that can make our work clearer. The suggested limits have been added (Please see lines 322-323 and 327-329).

Reviewer 2 Report

Comments and Suggestions for Authors

Lines 10–14 (Abstract):
The list of questionnaires used should clarify the purpose of each (e.g., Medi-Lite for MD adherence, FFQ for overall dietary habits). Consider briefly stating key findings on MD adherence to balance the emphasis on supplement use.

Line 33:
"increasing the risk of conditions such as obesity, diabetes and cardiovascular disease" – Consider citing more recent and adolescent-specific sources, or clarifying that these risks pertain to long-term outcomes.

Line 38–39:
Repetition of "excessive consumption of meat and processed products" – consider varying the language or combining with the previous sentence for smoother flow.

Line 46:
The phrase "peer social media" is unclear. Suggest revising to "peer influence, social media, or celebrity endorsements."

Line 52:
The repetition of “to the limited” appears to be a typographical error. Please revise to "contribute to the limited evidence on this topic."

Lines 59–60:
“During 2024, students attended six lessons on food education from.” — This sentence is incomplete. Please revise or complete the thought.

Line 80:
Since BMI is based on self-reported data, this limitation should be discussed in the discussion section. Accuracy of adolescent self-reporting can be questionable.

Lines 104–111:
Consider clarifying if any seasonal bias may have influenced food frequency reporting, especially for fruits and vegetables.

Lines 122–123:
The decision to exclude alcohol from the Medi-Lite score for this age group is reasonable but should be more explicitly stated in the methods, including whether the score was validated in this adjusted form.

Lines 141–143:
Consider reporting BMI categories (e.g., underweight, normal, overweight) rather than just the mean for greater interpretability.

Lines 167–172 (Results – Eating Habits):
You mention under- and over-consumption relative to guidelines. Please clarify whether the values reported are statistically significant deviations or merely descriptive.

Line 173:
A Medi-Lite score of 7.6 suggests low adherence, but the threshold ranges for low, medium, and high adherence should be briefly defined.

Line 195:
The figure legend for Figure 2 should indicate if differences are statistically significant.

Lines 209–215:
The discussion repeats much of the results. Suggest integrating results with external studies earlier to provide deeper interpretation rather than recapitulation.

Lines 229–234:
You attribute the higher rate of supplement use to the sport-oriented context. This is plausible, but please include a supporting citation or reference prior work with similar findings.

Line 241:
The distinction between team and individual sports influencing supplement use is interesting. Consider exploring whether coaching culture or access to trainers differs by sport type, potentially influencing these behaviors.

Line 247:
Typo: “could be face” should be “could face.”

Lines 253–257 (Limitations):
While limitations are mentioned, it would be beneficial to acknowledge potential confounding variables (e.g., socioeconomic status, dietary education at home) that were not measured but could influence results.

Lines 263–270 (Conclusion):
This paragraph is strong but could be improved by including a practical example of what a “targeted intervention” in schools might entail (e.g., curriculum changes, peer-led workshops, or digital apps for nutrition tracking).

Line 285:
The "Acknowledgments" section should be removed if there truly are no acknowledgments, or alternatively state “None.”

Author Response

Lines 10–14 (Abstract):

The list of questionnaires used should clarify the purpose of each (e.g., Medi-Lite for MD adherence, FFQ for overall dietary habits). Consider briefly stating key findings on MD adherence to balance the emphasis on supplement use.

R: Thank you for your helpful comment, which allowed us to improve the clarity of the abstract. We made changes accordingly to better reflect your suggestions.  (Please see lines 15-18)

Line 33:

"increasing the risk of conditions such as obesity, diabetes and cardiovascular disease" – Consider citing more recent and adolescent-specific sources, or clarifying that these risks pertain to long-term outcomes.

R: Thank you for your suggestion. We have included more recent references and revised the sentence to better clarify that these risks refer to long-term health outcomes. (Please see lines 57-59)

Line 38–39:

Repetition of "excessive consumption of meat and processed products" – consider varying the language or combining with the previous sentence for smoother flow.

R: In accordance with the suggestion, we revised the text to improve its flow and reduce repetition. (Please see line 63)

Line 46:

The phrase "peer social media" is unclear. Suggest revising to "peer influence, social media, or celebrity endorsements."

R: Thank you for pointing out this detail. We have changed the text as you specified so as to improve the clarity of the sentence. (Please see line 71)

Line 52:

The repetition of “to the limited” appears to be a typographical error. Please revise to "contribute to the limited evidence on this topic."

R: Thank you for pointing out the typographical error, which has been corrected in the revised version of the manuscript. (Please see lines 66)

Lines 59–60:

“During 2024, students attended six lessons on food education from.” — This sentence is incomplete. Please revise or complete the thought.

R: We appreciate your attention to this error, which has been addressed in the revised manuscript. (Please see line 76)

Line 80:

Since BMI is based on self-reported data, this limitation should be discussed in the discussion section. Accuracy of adolescent self-reporting can be questionable.

R: We thank you for bringing this aspect to our attention. In accordance with your suggestion, we have included this aspect among the limitations to be emphasized in the study. (Please see lines 327-329)

Lines 104–111:

Consider clarifying if any seasonal bias may have influenced food frequency reporting, especially for fruits and vegetables.

R: We agree that the consumption of certain food groups may vary, particularly due to seasonality, as is often the case with fruits and vegetables. For this reason, and in line with your suggestion, we have included this aspect as a limitation of the study. (Please see lines 329-385)

Lines 122–123:

The decision to exclude alcohol from the Medi-Lite score for this age group is reasonable but should be more explicitly stated in the methods, including whether the score was validated in this adjusted form.

R: We adapted the inclusion of alcohol consumption in the MediLite questionnaire based on previous studies that applied the same validated instrument to adolescent populations. The sale and supply of alcoholic beverages to people under the age of 18 is strictly prohibited in Italy. Therefore, the inclusion of alcohol intake in the assessment of adherence to the Mediterranean diet for this age group could introduce a substantial bias. In line with the established literature, we excluded alcohol to ensure the accuracy and validity of our results. However, we have now revised the Methods section to clarify this point more explicitly. (Please see lines 163-165)

Dinu, M.; Lotti, S.; Pagliai, G.; Pisciotta, L.; Zavatarelli, M.; Borriello, M.; Solinas, R.; Galuffo, R.; Clavarino, A.; Acerra, E.; et al. Mediterranean Diet Adherence in a Sample of Italian Adolescents Attending Secondary School—The “#facciamoComunicAzione” Project. Nutrients 2021, 13, 2806. https://doi.org/10.3390/nu13082806

Lines 141–143:

Consider reporting BMI categories (e.g., underweight, normal, overweight) rather than just the mean for greater interpretability.

R: Thank you very much for your valuable comment, which helped us improve the clarity and quality of our manuscript. Following your suggestion, we have included not only the mean BMI of the sample but also the distribution of participants across BMI categories (underweight, normal weight, overweight/obese) to enhance interpretability. (Please see lines 195-197)

Lines 167–172 (Results – Eating Habits):

You mention under- and over-consumption relative to guidelines. Please clarify whether the values reported are statistically significant deviations or merely descriptive.

R: Thank you very much for your insightful comment. The distinction between under- and over-consumption was made by comparing the reported dietary intake for each food group, as collected through the FFQ, with the recommended intake levels outlined in the current Italian dietary guidelines for adolescents aged 11 to 14 years. These comparisons are descriptive and based on how intake levels align with or deviate from the guidelines.

Line 173:

A Medi-Lite score of 7.6 suggests low adherence, but the threshold ranges for low, medium, and high adherence should be briefly defined.

R: The validated Medi-Lite questionnaire does not define specific threshold ranges to categorize adherence as low, medium, or high. However, the total score ranges from 0 (indicating minimal adherence) to 16 (indicating maximum adherence) in our version, where the alcohol component was excluded. Based on this scale, the mean score of 7.6 can be interpreted as reflecting a relatively low level of adherence to the Mediterranean diet.

Line 195:

The figure legend for Figure 2 should indicate if differences are statistically significant.

R: Following your suggestion, we have added information regarding the statistical significance of the differences directly in the legend of Figure 2 to improve clarity and interpretation. (Please see Line 257)

Lines 209–215:

The discussion repeats much of the results. Suggest integrating results with external studies earlier to provide deeper interpretation rather than recapitulation.

R: Thanks for the suggestion. We have lightened the initial part so that we start with the comparison with the present literature first (Please see lines 259-263).

Lines 229–234:

You attribute the higher rate of supplement use to the sport-oriented context. This is plausible, but please include a supporting citation or reference prior work with similar findings.

R: Thank you for your comment. We have added the requested references (Please see lines 291-302).

Line 241:

The distinction between team and individual sports influencing supplement use is interesting. Consider exploring whether coaching culture or access to trainers differs by sport type, potentially influencing these behaviors.

R: Thanks for the interesting insight, unfortunately we do not have this data available in our study, but we will take it into account for future ones.

Line 247:

Typo: “could be face” should be “could face.”

R: We appreciate your attention to this error, which has been addressed in the revised manuscript. (Please see line 316)

Lines 253–257 (Limitations):

While limitations are mentioned, it would be beneficial to acknowledge potential confounding variables (e.g., socioeconomic status, dietary education at home) that were not measured but could influence results.

R: Thank you for this valuable comment, which has helped improve the quality of our paper. For this reason, we have now included a statement acknowledging this limitation in the study. (Please see lines 385-387)

Lines 263–270 (Conclusion):

This paragraph is strong but could be improved by including a practical example of what a “targeted intervention” in schools might entail (e.g., curriculum changes, peer-led workshops, or digital apps for nutrition tracking).

R: To improve the conclusions of our manuscript and in response to your recommendation, we have specified targeted interventions such as nutrition education programs in schools or sports centers, better tailored to our study population. (Please see Line 398)

Line 285:

The "Acknowledgments" section should be removed if there truly are no acknowledgments, or alternatively state “None.”

R: Following your suggestion, we have specified “None” in the acknowledgments section. (Please see Line 416)

Round 2

Reviewer 1 Report

Comments and Suggestions for Authors

The authors have significantly revised and improved their manuscript.